# Functional characterization of extrinsic tongue muscles in the *Pink1-/-* rat model of Parkinson disease

Tiffany J. Glass[1]*, Cynthia A. Kelm-Nelson[1], John C. Szot[1], Jacob M. Lake[1], Nadine P. Connor[1,2], Michelle R. Ciucci[1,2]

1 Department of Surgery, University of Wisconsin, Madison, Wisconsin, United States of America,
2 Department of Communication Sciences and Disorders, University of Wisconsin, Madison, Wisconsin, United States of America

* glass@surgery.wisc.edu

**Data Availability Statement:** All relevant data are within the paper and its Supporting Information files.

## Abstract

Parkinson disease (PD) is associated with speech and swallowing difficulties likely due to pathology in widespread brain and nervous system regions. In post-mortem studies of PD, pathology has been reported in pharyngeal and laryngeal nerves and muscles. However, it is unknown whether PD is associated with neuromuscular changes in the tongue. Prior work in a rat model of PD (*Pink1-/-*) showed oromotor and swallowing deficits in the premanifest stage which suggested sensorimotor impairments of these functions. The present study tested the hypothesis that *Pink1-/-* rats show altered tongue function coinciding with neuromuscular differences within tongue muscles compared to wildtype (WT). Male *Pink1-/-* and WT rats underwent behavioral tongue function assays at 4 and 6 months of age (n = 7–8 rats per group), which are time points early in the disease. At 6 months, genioglossus (GG) and styloglossus (SG) muscles were analyzed for myosin heavy chain isoforms (MyHC), α-synuclein levels, myofiber size, centrally nucleated myofibers, and neuromuscular junction (NMJ) innervation. *Pink1-/-* showed greater tongue press force variability, and greater tongue press forces and rates as compared to WT. Additionally, *Pink1-/-* showed relative increases of MyHC 2a in SG, but typical MyHC profiles in GG. Western blots revealed *Pink1-/-* had more α-synuclein protein than WT in GG, but not in SG. There were no differences between *Pink1-/-* and WT in myofiber size, centrally-nucleated myofibers, or NMJ innervation. α-synuclein protein was observed in nerves, NMJ, and vessels in both genotypes. Findings at these early disease stages suggest small changes or no changes in several peripheral biological measures, and intact motor innervation of tongue muscles. Future work should evaluate these measures at later disease stages to determine when robust pathological peripheral change contributes to functional change, and what CNS deficits cause behavioral changes. Understanding how PD affects central and peripheral mechanisms will help determine therapy targets for speech and swallowing disorders.

**Funding:** This work was supported through the National Institute on Deafness and Other Communication Disorders (NIDCD) R01 DC014358 (M.R.C.), R01 DC018071 (N.P.C.), R01DC008149 (N.P.C.), R21 DC016135 (C.A.K.) (URL: https://www.nidcd.nih.gov/).

**Competing interests:** The authors have declared that no competing interests exist.

## Introduction

Parkinson disease (PD) involves widespread pathology in central and peripheral substrates [1, 2]. Although PD is classically defined by central dopamine loss, our understanding of the widespread dysfunction that causes motor, sensory, autonomic, sleep, mood, speech, and cognitive issues is evolving [3, 4]. Dysphagia, or difficulty swallowing, occurs in a high percentage of patients with PD [5]. Dysphagia can emerge early in PD, even in the premanifest (preclinical) stage, prior to the appearance of classic motor signs. Disruptions of swallowing can include delays in swallow initiation, vallecular and pharyngeal residue, and silent aspiration [6] which negatively impact quality of life. Swallowing involves highly coordinated movements of the tongue muscles, and functional studies of the oropharyngeal swallow in patients with PD have reported alterations of tongue movements during swallow, including altered tongue strength and altered timing of movements [7]. Unfortunately, dysphagia does not reliably respond to standard PD treatments including levodopa or deep brain stimulation [5, 6, 8].

Functional movement of the tongue requires complex coordination of intrinsic and extrinsic tongue muscles. The genioglossus (GG) and styloglossus (SG) muscles are both extrinsic tongue muscles. The GG muscle originates from the mandible and inserts along the ventral surface of the intrinsic tongue and the hyoid bone [9]. The SG originates from the styloid process of the temporal bone, and inserts into the lateral and inferior regions of the intrinsic tongue [9]. Although these two muscles share embryonic origins and innervation [10, 11], they have opposing actions. While the GG acts to protrude the tongue, the SG acts to pull the tongue backward. When activated together, the GG and SG may help to stabilize the bolus pathway. Therefore, both muscles are critical to tongue movements involved in deglutition [12, 13].

Prior histopathological investigations of PD have shown the presence of peripheral neuromuscular pathology of pharyngeal muscles involved in swallowing in humans with late-stage PD. These changes include alterations in myofiber size and myosin heavy chain isoform profiles (MyHC), increases in incidence of myofibers with centralized nuclei, and the presence of α-synuclein aggregates localized specifically to neuromuscular junctions (NMJ) and peripheral nerves [14–17]. These findings raise the possibility that peripheral neuromuscular differences involving muscles of swallowing may coincide with swallowing deficits in PD. However, it is unknown whether neuromuscular differences of oropharyngeal muscles occur in patients at pre-manifest or early stages of PD, when swallowing deficits begin to emerge. While logistical and feasibility barriers preclude studies of peripheral neuropathology in humans at pre-manifest stages of PD, genetic animal models of PD provide opportunities to examine these questions in controlled studies. With genetic animal models, early disease processes and biological differences can be studied with a precision that is not feasible in humans. Using these models, we can study early changes in sensorimotor function of the tongue that may coincide with significant changes in neuromuscular biology of the tongue muscles [18].

The PTEN-induced putative kinase 1 knockout (*Pink1-/-*) rat model of early-onset PD demonstrates CNS pathology including loss of dopaminergic neurons in the substantia nigra, α-synuclein deposition in the nucleus ambiguus and periaqueductal gray, and significant perturbations of oromotor function and swallowing performance that precede the emergence of classic gross-motor signs of PD [19–21]. Compared to age-matched controls, *Pink1-/-* rats in early stages of disease show behavioral differences in tongue function that include significantly increased tongue press forces, and increased tongue press variability (initial overshooting and then decay of function) during a licking task [21]. Force and timing variabilities are implicated in hallmarks of PD as seen in limb movement [22, 23] and have been previously shown to occur in swallowing differences that emerge early in PD [24]. *Pink1-/-* rats also show

reductions in mastication rates, increases in food bolus size during swallowing, increases in average bolus velocity during swallowing [20], and increased variability in intervals between bites when consuming dry pasta [21]. The causes of oromotor dysfunction underlying dysphagia in PD are mostly unknown but do include decreased norepinephrine in the locus coeruleus and increased insoluble a-synuclein in brainstem regions important for tongue use and deglutition. We hypothesized that oromotor deficits in the *Pink1 -/-* model may include biological differences of muscles involved in deglutition. In the present study, the *Pink1 -/-* rat model of PD was used to test the hypothesis that neuromuscular differences occur in the tongue muscles at 6 months of age. At 6 months of age in this model, oromotor functions are significantly impacted, but more classic gross motor signs of the disease have yet to emerge [21].

## Methods

### Animals

This study was approved by the University of Wisconsin-Madison School of Medicine and Public Health Institutional Animal Care and Use Committee (IACUC) (Protocol numbers: M005486 and M005177), and adhered to Guide for the Care and Use of Laboratory Animals [25]. Male Long-Evans rats obtained from SAGE™ Research Labs, (Boyertown, PA, USA), were housed in same-genotype pairs on a reversed 12:12 hour light:dark cycle. Experimental groups comprised of 7 *Pink1 -/-* rats and 8 WT rats euthanized at 4 months of age, and 8 *Pink1 -/-* rats and 7 WT rats euthanized at 6 months of age. Animals were weighed to confirm weights compatible with good health prior to testing and euthanasia. The body weight means (SD) were: WT 4 month old: 364.6 g (33.49), *Pink1 -/-* 4 month old: 471.6 g (8.64), WT 6 month old: 397.1 g (18.56), *Pink1 -/-* 6 month old: 472.5 g (61.57). Incidental animal death (one *Pink1 -/-* rat in the 4 month cohort, and one WT rat in the 6 month cohort) caused unbalanced group numbers. However, a prior study indicates these group sizes are sufficient for detection of significant differences in muscle biology in this experimental system [26]. Worker bias was mitigated through the use of an alphanumeric code to identify each animal in the study. Separately, for the purpose of NMJ analysis validation, this study used muscle samples obtained from two surgically denervated male CD® (Sprague Dawley) IGS rats.

### Tongue press measurements

Behavioral assays were conducted during the dark period of the light cycle using partial red illumination. A tongue press paradigm was used as previously described [27–29], in which rats elicited a water reward by using their tongue to press a disk linked to a force transducer. Rats were gradually trained to lick and press the disk with their tongue to elicit a water reward (Fig 1). In the first week, rats were acclimated to handling. In the second week, a gradual water regulation process occurred in which access to water was regulated for progressively longer periods of time each day prior to interaction with the testing apparatus, to a final regulation period

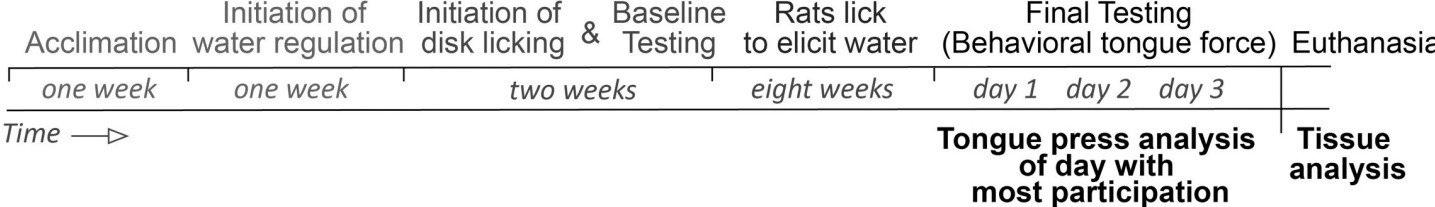

**Fig 1. Depiction of behavioral training and testing periods for collection of tongue press force data and tongue press timing data.**

of freely available water for three hours per day. In the subsequent two weeks, rats learned to lick the disk, which is a process that exerts at least 0.2 g of force, to elicit an automatically dispensed water reward, and underwent testing for baseline tongue press performance. This was followed by eight weeks during which rats licked the disk to obtain water rewards, but were not required to exert appreciable or increasing forces with their tongue during the process. During this period, rats were removed from the testing apparatus immediately after initial demonstration of behavioral compliance with the licking procedure. At the end of 8 weeks, rats underwent three days of final increment testing, in which tongue force targets were progressively increased, requiring progressively greater tongue forces to elicit water rewards. To maximize motivation, these force thresholds were also coupled with a randomized water dispense function, such that between one and five tongue press events were required to elicit each water reward (variable ratio 5 schedule). During testing sessions, behavioral tongue force pressure and timing data were collected. The Estimated Maximum Press (EMP) force was the average of the top 10 highest presses across the three testing days for each rat. In addition to force, the timepoint of each tongue press event was automatically detected, which permitted analysis of tongue press behavior over time during the testing session. On occasion, as previously described [27], a few rats may begin to use their front incisors, rather than their tongues, to press the disk to elicit water rewards. Staff monitored rats closely during testing and rats that perseverated in this behavior were removed from the testing apparatus, and press force values attributable to presses with incisors were manually removed from data sets. For each rat, participation in testing was quantified for each of the three test days, and the test day with the highest participation level was identified. Tongue press timing and press forces were analyzed for each rat using data from the final testing day that demonstrated the greatest participation. To ensure hydration throughout this process, rats were additionally provided with freely available water for three hours each day after the completion of every session. The conclusion of the final testing and subsequent euthanasia coincided with endpoints of either 4 months of age or 6 months of age.

### Surgically denervated muscles for biological validation of neuromuscular junction analysis

Unilateral surgical denervation provided the means to obtain control muscle tissue in which an absence of functional innervation was known, thereby enhancing the rigor of assays quantifying denervation. These biological control muscles were obtained from WT rats that were not otherwise involved in experiments incorporating *Pink1 -/-* rats. WT rats were deeply anesthetized through isoflurane inhalation and were administered sustained release buprenorphine for analgesia. Following medial reflection of the digastric muscle, the hypoglossal nerve was bisected unilaterally (either only on the right or only on the left side). The transversus mandibular muscle was reflected rostrally, the mylohyoid muscle was reflected caudally, the geniohyoid muscle was bluntly separated, and the lingual nerve was bisected unilaterally. Following unilateral nerve cuts, skin incisions were sutured closed, and monitoring with postoperative analgesia was maintained daily for three to five days, with daily monitoring five times/week thereafter. Three weeks later, rats were euthanized, and extrinsic tongue muscles were isolated for analysis.

### Muscle isolation and sample handling

Prior to sample isolation, rats were deeply anesthetized through isoflurane inhalation and pentobarbital, and ultimately euthanized through an overdose of Beuthanasia-D. Rats were placed in a recumbent position, and extrinsic tongue muscles were excised through a ventral

approach. Muscles used for protein analysis were transferred to Eppendorf tubes and promptly frozen in liquid nitrogen. Muscles used for tissue sectioning were promptly embedded in Optimum Cutting Temperature (O.C.T.) Compound in a plastic sample cassette, which was frozen through immersion in isopentane that had been pre-chilled in liquid nitrogen. All samples were stored at -80˚ C prior to analysis.

## Protein isolation

Muscles were minced with small scissors in RIPA buffer containing phosphatase and protease inhibitors, sonicated, incubated on ice for 30 minutes, then centrifuged at 6000 rpm for 15 minutes. The resulting supernatant was reserved for western blot analysis of α-synuclein as described below, and the resulting pellet was processed for MyHC profile analysis as described below.

## Western blot methods

Protein supernatant (50 μg of total protein) was mixed with 2X Laemmli buffer (Bio-Rad Laboratories, Hercules, CA, USA) with 2-mercaptoethanol, denatured at 95˚C for 5 min, and lysates were resolved on a Criterion Precast Gel (4–20% gradient Tris HCL-polyacrylamide gels,1.0mm, 12 x 2 Well Comb, Bio Rad Laboratories). Pre-stained protein standards (Precision Plus Protein Dual Xtra Standards, Bio Rad Laboratories) were also included on gels as molecular mass markers. Commercial mouse brain lysate (10 μg total protein; Cell Signaling Technology, Inc., Danvers, MA, USA) was also run on each gel as a positive and technical control. Gels were subjected to electrophoresis in 10X Tris-buffered saline buffer with glycine (TBS, Bio Rad) for 1:15 h at 125 V then transferred in 10X TBS with glycine (Bio Rad) with 20% methanol for 1.5 h at 100 V onto 0.2 μm nitrocellulose membranes (Bio Rad Laboratories). Membranes were blocked with filtered 5% non-fat milk in Tris-buffered saline with 0.1% Tween-20 (TBS-T) for 1 h at 4˚C with agitation. Blots were probed with primary antibodies for anti-alpha synuclein (1:250, #2642S, Cell Signaling Technology, Inc.) and loading control (anti-β actin, 1:40,000, Millipore, Billerica, MA, USA) overnight (20.5 h) at 4˚C with constant agitation.

Following primary antibody incubation, blots were washed, then probed with horseradish peroxidase-conjugated anti-rabbit IgG (1:5,000 dilution, Cell Signaling Technology Inc.) and anti-mouse IgG (1:10,000 dilution, Cell Signaling Technology Inc.). Blots were washed in TBS-T and enhanced chemiluminescence substrate with Super Signal West Pico (Thermo Scientific, Madison, WI, USA). A ChemiDoc-IT2 Imager (UVP, LLC, Cambridge, UK) was used to detect and capture images. The grayscale band at 18 kDa was analyzed with ImageJ (National Institutes of Health); the density was normalized to β actin internal controls (43 kDa). For immunoblotting, a Gel Analysis method outlined in the ImageJ documentation was used: http://rsb.info.nih.gov/ij/docs/menus/analyze.html#gels.

## Myosin Heavy Chain (MyHC) isoform profiles

Sample pellets were processed using reagents previously described [30]. Pellets were briefly sonicated in protein extraction buffer, incubated at 4˚C for 1 hour, centrifuged for 40 min at 14.5 rpm, and resulting supernatant was analyzed for protein concentration through a Bradford Protein Assay. Protein (400 ng per well) was separated on a large format 6% acrylamide/30% glycerol separating gel, 4% acrylamide/30%glycerol stacking gel run for 24 hours, silver stained, and analyzed by UN-SCAN-IT gel analysis software (Silk Scientific). The software segment analysis tools were used to determine the average pixel value for each MyHC band. The average pixel value for each band was used to calculate the relative percentage each MyHC

isoform contributed to the total MyHC isoform protein in each sample lane, as previously described [26, 31].

## Immunofluorescence staining

**Staining for myofiber analysis.**   Genioglossus and styloglossus muscles were embedded in OCT and frozen in isopentane pre-cooled in liquid nitrogen. For myofiber size analyses, thin sections (10 μm) were fixed on slides for 10 minutes in 4% paraformaldehyde (PFA), rinsed, blocked, and incubated overnight at 4˚C in primary antibodies for Laminin gamma-1 (D18; DSHB, applied at 1:100) and Neural Cell Adhesion Molecule (NCAM) (Millipore, AB5032, applied at 1:250) [32]. After rinses and application of the secondary antibodies AF488 (1:800) and AF594 (1:500) for 1 hour at room temperature, slides were rinsed, mounted with DAPI, and imaged. For each staining iteration, negative control slides were prepared with primary antibodies omitted. Biological positive controls for muscle pathology consisted of concurrent staining of muscle tissue sectioned from a rat model of amyotrophic lateral sclerosis.

**Staining for Neuromuscular Junction (NMJ) analysis and α-synuclein localization.** NMJ staining and analysis methods were based on prior reports of denervation processes in murine models of amyotrophic lateral sclerosis [33, 34]. Thin sections (10 μm) were fixed on slides for 15 minutes in 4% PFA, rinsed, blocked, and incubated overnight in the primary antibodies rabbit anti-α-synuclein (1:400, Cell SignalingTechnology D37A6), mouse anti-synaptotagmin 2 (1:8, DSHB, ZNP-1), and chicken anti-neurofilament-M (NF-M) (1:1000, antibodies online ABIN361355) to detect pre-synaptic structures, and alpha Bungarotoxin-AF488 (1:1000, Invitrogen) to detect motor endplates through labeling of acetylcholine receptors. Secondary antibodies used were anti-chicken Cy3 (1:800), anti-mouse AF568 (1:500), and anti-rabbit AF633 (1:250). Thin sections of rat hippocampus were used as biological positive controls to verify detection of α-synuclein, and signal specificity for α-synuclein was verified through the use of a rabbit isotype control, as well as sections in which the primary antibodies were omitted. Biological positive control tissues of surgically denervated extrinsic tongue muscles were processed concurrently to verify specificity and sensitivity of NMJ innervation staining, image acquisition settings, and analysis.

**Image acquisition and analysis.**   Images were acquired with an Olympus BX53 Upright Microscope fitted with a DP80 Dual CCD Color and Monochrome Camera, Prior XYZ Motorized Stage Kit and cellSens Dimensions software (Olympus). Image acquisition exposure settings were optimized during each imaging session using a combination of experimental slides and control slides.

*Imaging and analysis of myofibers.* Programmed, automated image stitching permitted multiple fields-of-view to be photographed with a 40x objective then digitally combined to capture one tissue cross-section of each GG and SG muscle taken per animal. Tissue section composite images were analyzed through the semi-automated MatLab application SMASH [35] to determine myofiber cross-sectional area (CSA) and total number of analyzable myofibers per tissue section. Separately, images were manually analyzed in Adobe Photoshop CC, using the Count tool and Channel tool, to identify myofibers with centralized nuclei and myofibers staining positive for NCAM.

*Imaging and analysis of NMJ.* Between 50 and 75 NMJ from each muscle, distributed across several tissue sections, were photographed using a 20x objective. Multichannel images were manually analyzed in Adobe Photoshop CS to rate each NMJ as either innervated (co-localization of pre-synaptic signal at motor endplates) or denervated (an absence of pre-synaptic signal at motor endplates), by alternately viewing single-channel signal and merged multichannel signal for each NMJ. Manual analysis was also used to rate each NMJ and as either

positive or negative for co-localization of strong α-synuclein signal. Analysis was performed by a single rater. Intra-rater reproducibility was evaluated through re-rating of all images used for approximately 30% of the experimental samples. Samples to be re-rated were randomly selected from each of the experimental groups. Results of a Spearman correlation indicated good intra-rater reliability for evaluation of NMJ innervation ($r_s$ = 0.81, p = 0.03), and good intra-rater reliability for manual evaluation of strong α-synuclein signal at the NMJ ($r_s$ = 0.89, p = 0.005).

**Statistics.** Tongue press force and tongue press force variation were analyzed for main effects of two early timepoints in disease progression (4 months and 6 months) and genotype (WT and *Pink1 -/-*) through two-way analysis of variance (ANOVA) with Tukey's post-hoc tests, using GraphPad Prism v. 7.04 and v. 8.4.3. For ANOVAs, adherence to assumptions of normality and variance were confirmed through the Shapiro-Wilk test of normality and Levene's test of equality of variances. Tongue press timing data and tongue press force data over the duration of the testing session were analyzed for main effects of time (in 30 second intervals through the duration of the testing session) and genotype (WT and *Pink1 -/-*), as well as the interaction between time and genotype, with linear mixed effect models, using SAS software (version 9.4, SAS Institute Inc., Cary, NC). Tongue press timing data and force data were compared between genotypes at each time point using Wilcoxon Rank Sum Tests. Biological muscle measures were analyzed through one-tailed or two-tailed t-tests in GraphPad Prism v. 7.04 and v. 8.4.3 or SigmaPlot v. 13.0 (SysStat Software). Significance was set at alpha = 0.05.

## Results

### Behavioral tongue press measures

Compared to WT, *Pink1-/-* groups showed significantly greater maximum forces of tongue presses, significantly greater variability in tongue press force, and significantly greater frequencies, or rates, of tongue presses. Analysis of maximum tongue forces showed significant main effects for both genotype (p<0.0001) and age (p = 0.04), in the absence of significant interaction between genotype and age (p = 0.06). Post-hoc testing revealed significantly increased forces generated by *Pink1-/-* relative to WT at both 4 months of age (p<0.0001), and 6 months of age (p<0.0001). *Pink1-/-* also showed significantly greater tongue press forces at 6 months of age relative to *Pink1-/-* at 4 months of age (p = 0.04) (Fig 2A). Analysis of tongue press force variation indicated a significant main effect for genotype (p<0.0001). Post-hoc testing revealed *Pink1-/-* had greater force variability compared to WT at both 4 months of age (p<0.0001) and 6 months of age (p<0.0001) (Fig 2B).

Analysis of tongue press rates across 30-second timepoint increments at 4 months of age showed significant main effects for genotype (p = .007) and timepoint (p < .0001) with no significant interaction between test session timepoint and genotype. Genotype comparisons at discrete timepoints through Wilcoxon Rank Sum Tests indicated significant differences between genotypes at some timepoints prior to 150 s of testing (Fig 2C). Similarly, analysis of tongue press rates across 30-second timepoint increments at 6 months of age showed significant main effects for genotype (p = .0009) and timepoint (p < .0001) with no significant interaction between test session timepoint and genotype. There were significant differences between genotypes at some timepoints prior to 120 seconds of testing (Fig 2D). Analysis of tongue press force across 30-second timepoint increments at 4 months of age showed significant main effects for genotype (p < .0001) and timepoint (p = .005) with no significant interaction between test session timepoint and genotype. There were significant differences between genotypes at some timepoints prior to 210 seconds of testing (Fig 2E). Similarly, analysis of tongue press force across 30-second timepoint increments at 6 months of age showed

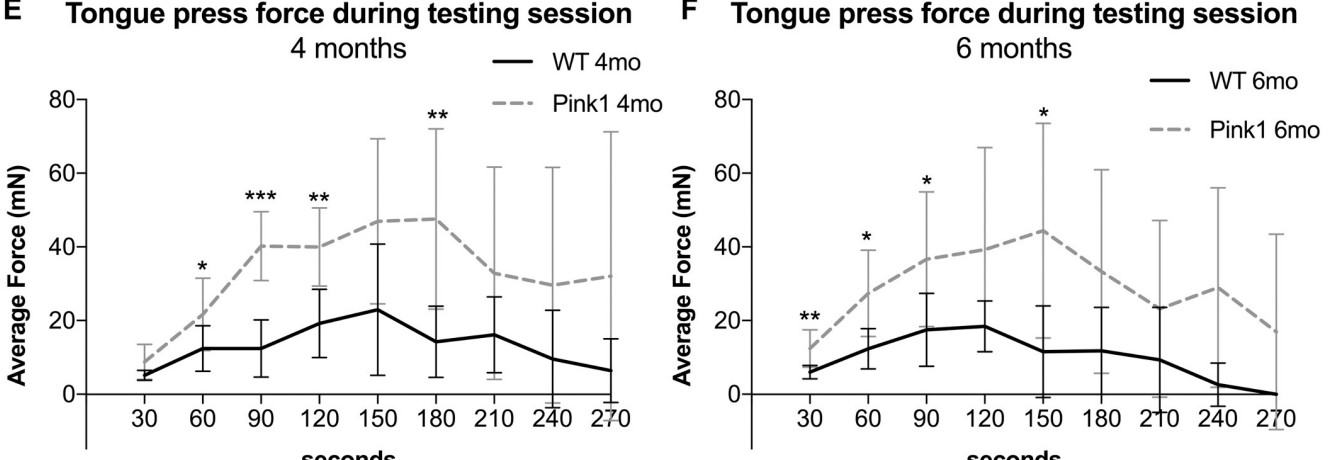

**Fig 2. *Pink1-/-* rats exert behavioral tongue pressures of greater force, greater force variability, and with greater press frequency than WT control rats.** A) The Estimated Maximum Press (EMP) force was the average of the top 10 highest presses from each of the three testing days for each rat. B) Coefficient of variation of tongue forces across the duration of the testing session day that had the highest degree of rat participation. C) The number of tongue presses per second, calculated in 30 second intervals, across the duration of the testing session day, for rats at 4 months of age. Starting N = 7–8 per group. D) The number

of tongue presses per second across the duration of testing session for rats at 6 months of age. Starting N = 7–8 per group. E) The average tongue press force, calculated in 30 second intervals, across the duration of the testing session day, for rats at 4 months of age. Starting N = 7–8 per group. F) The average tongue press force across the duration of testing session for rats at 6 months of age. Starting N = 7–8 per group.

significant main effects for genotype (p = .001) and timepoint (p = .045) with no significant interaction between test session timepoint and genotype. There were significant differences between genotypes at some timepoints prior to 180 s of testing (Fig 2F). Due to maladaptive behavior that emerged after initial compliance, 1–2 rats per group were removed from the testing chamber early, prior to the 270 second timepoint. The 6-month timepoint was selected for subsequent biochemical and immunohistochemical investigation of extrinsic tongue muscle biology.

### Measures of extrinsic tongue muscle structure

At 6 months of age, *Pink1-/-* GG muscles showed a non-significant decrease in MyHC 2b relative to WT controls (p = 0.06), whereas *Pink1 -/-* SG showed significant increases of relative levels of MyHC 2a (p = 0.02). These findings are compatible with the presence of a fast-to-slow MyHC transition in *Pink1-/-* tongue muscles (Fig 3). Myofiber size was not significantly different between genotypes for either GG muscles, or SG muscles. Similarly, incidence of myofibers with centralized nuclei did not differ between genotypes for either GG or SG muscle. NCAM staining for all muscles suggested trivial incidence (below 1%) of myofibers positive for NCAM expression for both SG and GG of both genotypes (Fig 4).

### Extrinsic tongue muscle α-synuclein content and NMJ innervation

Western blot quantification of α-synuclein content in the extrinsic tongue muscles revealed significant increases in the amount of α-synuclein in GG (p = .04) but not SG (p = .46) (Fig 5). Because the muscle samples in their entirety were homogenized for western blot analysis, this assay did not provide information about the microanatomical structures with which α-synuclein is associated in these muscles. Therefore, immunofluorescence microscopy was used to discern microanatomical locations of α-synuclein. Quantification of photomicrographs revealed no differences between genotypes in α-synuclein content at NMJs (Fig 5B). Photomicrographs further confirmed the presence of α-synuclein in extrinsic tongue muscles of both WT and *Pink1-/-* rats within peripheral nerves, vessels, and NMJ in extrinsic tongue muscles (Figs 5 and 6), however α-synuclein distribution and prevalence was qualitatively collectively similar between genotype groups, and showed high individual variability. Because prior research suggested that NMJ could be disrupted in PD, NMJ innervation was analyzed in both genotypes. NMJ analysis of GG and SG revealed both genotype groups showed incidence of innervation within normal ranges for this assay [33], with no significant differences between genotype groups (Fig 6).

### Discussion

Prior studies have reported that dysphagia in late stages of PD coincides with pathological myofiber type grouping, evidence of myofiber degeneration, and pathology in NMJs [15, 17, 36, 37]. The present study characterized differences of tongue function in the *Pink1-/-* genetic rat model of PD at 4 and 6 months of age. Present findings confirmed prior reports that compared to WT controls, *Pink1-/-* rats at early disease stages exert significantly greater tongue forces, with greater overall force variability, and with greater tongue press rates [21]. Additionally, the present study evaluated multiple measures of muscle biology in extrinsic tongue

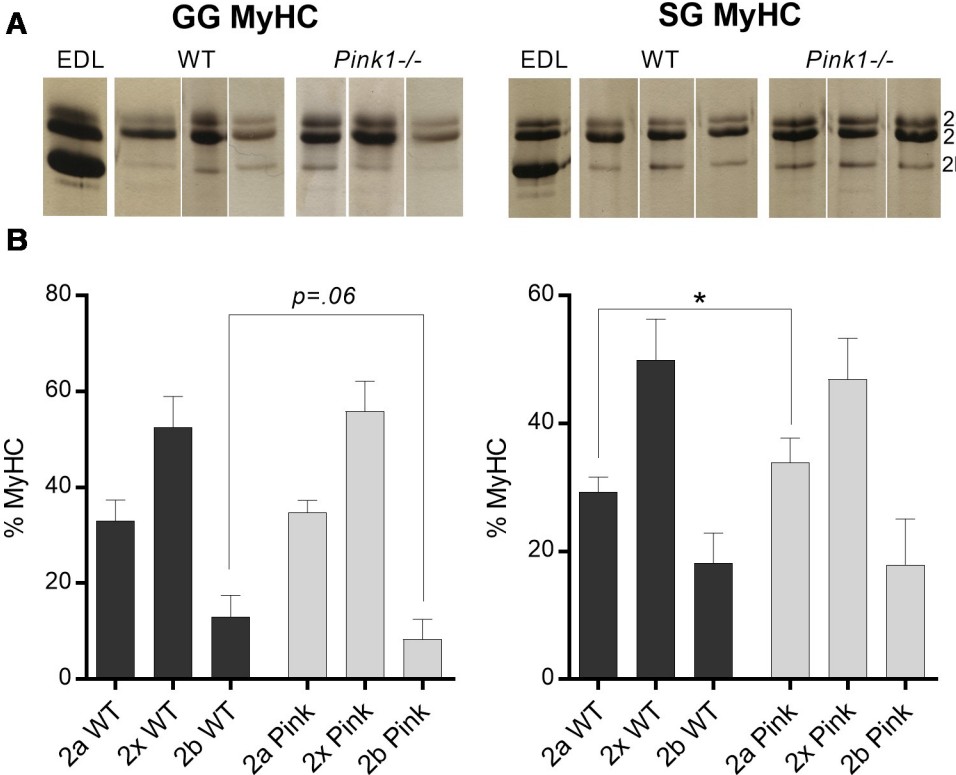

**Fig 3. Pink1 -/- rats have indications of fast-to-slow MyHC isoform transitions in the extrinsic tongue muscles.** A) Silver-stained MyHC bands of genioglossus (GG) and styloglossus (SG) muscle, alongside an extensor digitorum longus (EDL) muscle used as a control. Samples were identified through alphanumeric codes and run and analyzed in arbitrary order by workers blinded to sample identity. Lanes shown were compiled from different gels. **B)** Relative percentages of MyHC isoforms in GG and SG muscles. N = 7–8 per group.

muscles that are predominantly responsible for tongue protrusion and retrusion; testing the hypothesis that neuromuscular differences occur in extrinsic tongue muscles in the *Pink1-/-* model at an early timepoint coinciding with significant oromotor dysfunction. Findings overall were not compatible with an interpretation of robust neuromuscular pathology. Myofiber size, incidence of myofibers with centralized nuclei, prevalence of α-synuclein in the NMJ, and incidence of NMJ innervation were all within similar ranges for both WT and *Pink1-/-* genotype groups. While it was not hypothesized that the SG and GG muscles would be impacted differently in the *Pink1-/-* model, findings also included evidence of some biological differences between these muscles, such as significant fast-to-slow myofiber transition in SG muscles, but not GG muscles. However, the majority of muscle biology assays did not corroborate conclusions of robust biological distinctions between these muscles in the degree to which they were influenced by genotype. This study was designed to identify biological muscle differences that coincide with the presence of oromotor differences, but did not demonstrate mechanistic or causal relationships between statistically significant muscle outcomes and oromotor function. Prior reports of altered peripheral neuromuscular biology associated with PD have focused on late stages of PD. In contrast, findings of the present study suggest that oromotor differences at early stages of PD do not coincide with overt hallmarks of neuromuscular pathology in extrinsic tongue muscles.

Pathology of α-synuclein has been reported to occur in peripheral nerves, NMJ, and muscles of the larynx and pharynx at late stages of PD [14–16]. While α-synuclein is present in

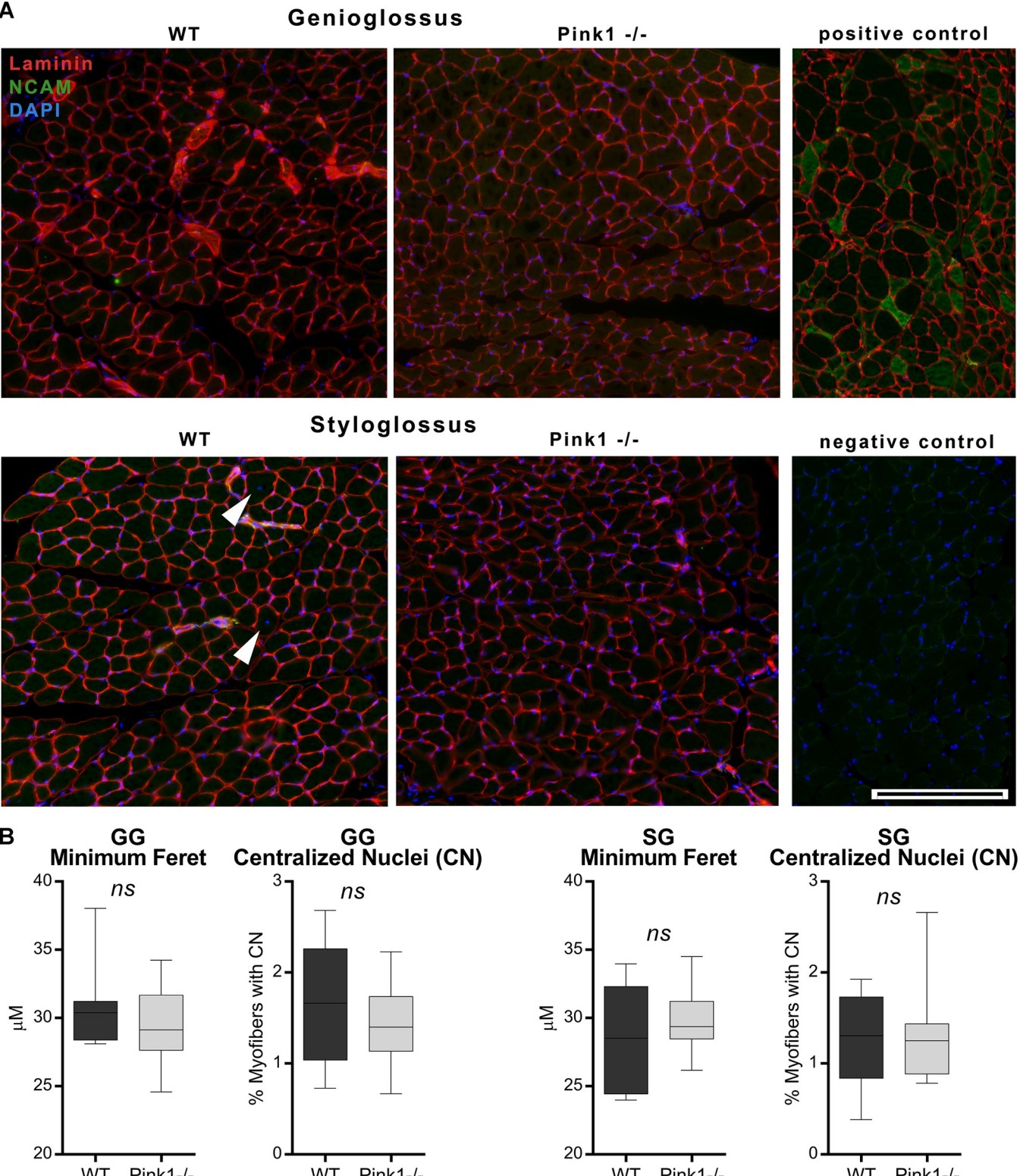

**Fig 4. Pink1 -/- extrinsic tongue muscles do not show altered myofiber size or increased incidence of centralized nuclei.** A) Images of genioglossus and styloglossus muscles showing myofiber borders through the D18 antibody (red), NCAM (green) and nuclei (blue). White arrowheads indicate myofibers with centralized nuclei (CN). Biological positive staining controls for NCAM comprised of limb muscle tissue from a rat model of ALS. Technical negative staining controls comprised of omission of either the primary antibodies, or secondary antibodies. Scale: 200 μm. B) Quantification of genioglossus (GG) and

styloglossus (SG) microscopy images showing no significant differences between groups in myofiber size or incidence of centrally nucleated myofibers. N = 7–8 per group.

cutaneous nerve fibers of both healthy controls and patients with PD, patients with PD have shown altered levels of α-synuclein in these nerves relative to controls [38, 39], and pathological α-synuclein aggregates have been reported in pharyngeal NMJs [15]. Further, the plausibility of increased risks for NMJ denervation in *Pink1-/-* models has been suggested by prior

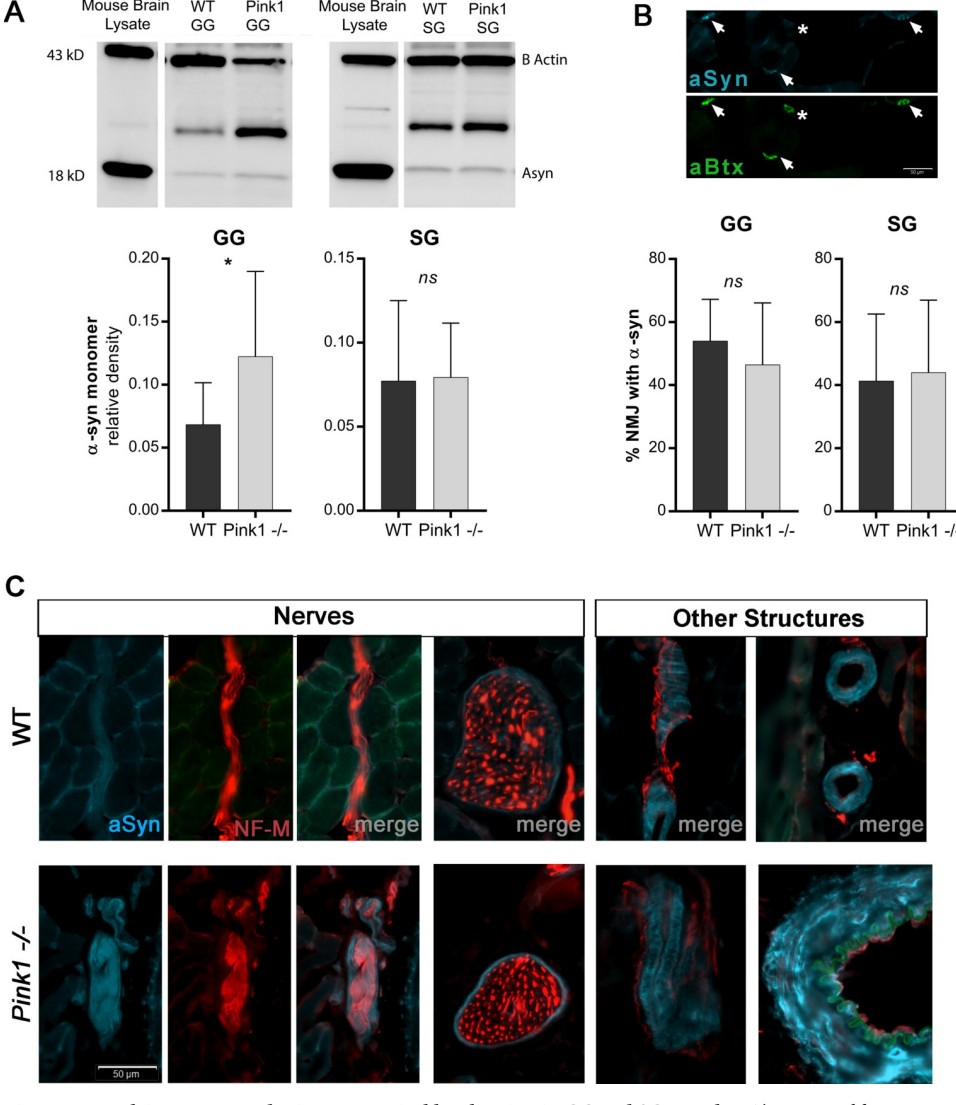

**Fig 5. α-synuclein content and microanatomical localization in GG and SG muscles. A)** Western blot quantification of α-synuclein in the GG and SG muscles of 6-month old rats. N = 7–8 per group. **B)** Immunofluorescence staining to quantify relative prevalence of of α-synuclein (cyan) at motor endplates of neuromuscular junctions (NMJ) (green) in thin tissue sections. Arrowheads indicate endplates with strong α-synuclein positivity, and asterisk indicates an endplate without strong α-synuclein signal. Prevalence of α-synuclein in NMJs showed no differences between genotype groups. Scale: 50 μm. **C)** Immunofluorescence staining revealed localization of α-synuclein to other microanatomical structures in extrinsic tongue muscles. Selected images representative of incidental observation of α-synuclein localized to regions of peripheral nerves (NF-M; red), and other structures in both WT and *Pink1 -/-* GG and SG. Scale: 50 μm.

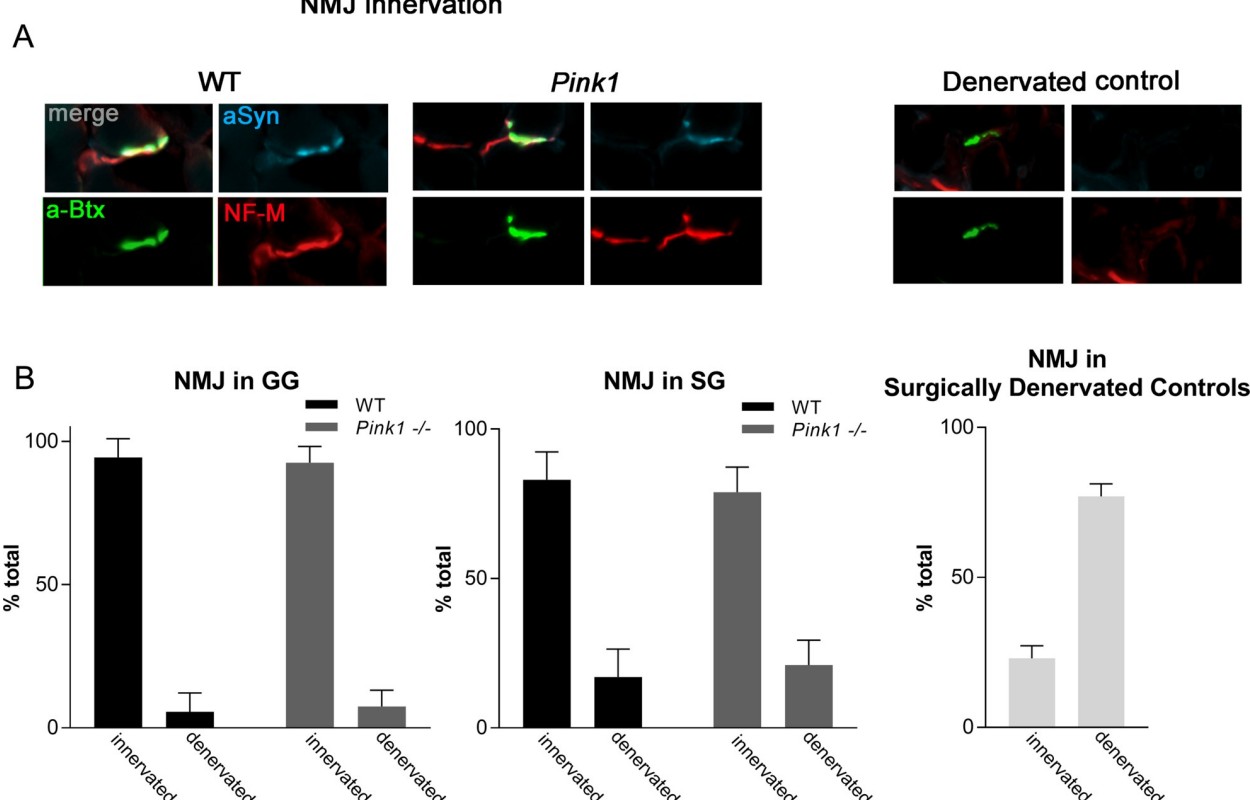

**Fig 6. Neuromuscular Junction (NMJ) analysis of extrinsic tongue muscles.** A) Immunofluorescence staining showing localization of α-synuclein (cyan) to neuromuscular junctions in extrinsic tongue muscles. Presynaptic structures are labeled with NF-M & ZNP-1 (red), and the post-synaptic motor endplate is labeled with alpha-bungarotoxin (green). Quantification of co-localization between green signal and bright red signal indicates intact NMJ innervation in both WT and *Pink1 -/-*. Biological control muscles in which rats had been surgically denervated confirmed the biological fidelity of this assay. B) Quantification of NMJ innervation in 6 month old *Pink1-/-* rats. N = 6 rats per group, 50–70 NMJs analyzed per rat. Surgically denervated SG controls provided biological and technical assay validation. N = 2 rats per group, 50–75 NMJs analyzed per rat.

work demonstrating denervation of the diaphragm in double knockout mice with loss of function in both Pink1 and Parkin [40]. In the present study we found significant increases in total α-synuclein protein in the GG of *Pink1-/-* group as compared to WT as indicated by western blot. However, microscopy analysis indicated no difference between *Pink1-/-* and WT in relative α-synuclein content in NMJs of the extrinsic tongue muscles, and no evidence of denervation of these muscles at the level of the NMJ. Alternative microanatomical localizations of α-synuclein associated with the tongue tissue that may contribute to total α-synuclein content measures may include salivary structures [41], red blood cells which may be present in varying proportions in tissue samples [42], and vascular endothelium [43]. These sources of α-synuclein may potentially explain findings of increased α-synuclein in *Pink1-/-* GG as evaluated by western blot, but not as evaluated by microscopy at the NMJ.

The presence of behaviorally altered tongue function in the absence of extensive or overt biological tongue muscle differences is compatible with the possibility that sensory dysfunction and central mechanisms play predominant roles in alteration of these oromotor functions in *Pink1-/-* at early disease timepoints. It has been suggested that in humans with PD, sensory impairments contribute to clinical deficits in speech and swallowing [44], and it is possible that mechanosensory impairments may emerge prior to chemosensory impairments [45].

Prior work in humans with PD has reported alterations in sensorimotor aspects of oral functions including deficits in jaw proprioception, deficits of tactile localization of sensation on the tongue, and deficits in head movement in response to oral sensation [46]. Reductions in mechanosensation of the base of tongue have also been reported to occur in humans with PD [45]. Successful completion of the tongue press task in the present study required fine control of tongue positioning and appropriate grading of tongue pressures exerted on an external object. In this experimental tongue press task, the use of the tongue to perceive the location and resistance of a metal disk and to detect an automated dispense of a water reward provided the means through which rats obtained information about appropriate tongue pressures and press frequencies required for the task. The *Pink1-/-* performance characteristics of tongue press forces and tongue press frequencies could be described as 'overshooting,' in that they showed a significant excess of the force and frequency targets that successful completion of the task demanded. Analysis in the present study indicated overshooting at the beginning of the task for the tongue press timing measure, while *Pink1 -/-* overshooting in tongue press force was somewhat more distributed throughout the duration of the test session; albeit with high individual variability. Analysis in a prior study of *Pink1 -/-* [21] indicated this overshooting in both tongue press timing and tongue press force occurred at the beginning of the task, but was not sustained for the duration of the task [21]. Experimental design differences between the two studies included differences in the lengths of time rats received exposure to the tongue press apparatus prior to final testing days. This notwithstanding, prior work has indicated humans with PD show lower endurance of the tongue specifically in a tongue press task [47], which may suggest parallels in some aspects of oromotor function of this rat model of PD and PD in humans.

Within-session slowing of tongue movements may be speculated to be attributable to a dopamine mechanism, as previous work with haloperidol [28] and 6-OHDA [27] shows that blocking D2 dopamine receptors and lesioning the medial forebrain bundle unilaterally both lead to decreased tongue forces and lick rates. However, earlier studies have indicated that *Pink1-/-* rats likely do not have significant dopamine depletion at the ages studied here [19, 48]. Alternatively, prior studies within this model show reduced tyrosine hydroxylase immunoreactive (TH-ir) staining in the locus coeruleus (LC) [20], as well as increased alpha synuclein in dopaminergic and noradrergic-mediated brainstem structures [20, 21, 49].

The water reward associated with the tongue press task used in this study necessitates coordination between licking and swallowing. An earlier study of swallowing differences in *Pink1-/-* reported increased bolus size and increased bolus speeds [20], which are swallowing characteristics compatible with impaired oropharyngeal sensation. The possibility of central causes of oromotor dysfunction has been suggested in prior reports of significantly reduced numbers of TH-ir cells in the LC of *Pink1 -/-* at 8 months of age, as well as a negative correlation between the numbers of TH-ir cells in the LC and the behavioral tongue function measures of tongue press force and tongue press variability [21]. The presence of these central differences, coinciding with only modest differences of peripheral extrinsic tongue muscle biology found in the present study, suggests that examination of peripheral and central structures associated with mechanosensation and sensorimotor integration will be of value in future work examining the physical underpinnings of oromotor dysfunction in this model of PD.

The absence of concurrent evaluation of central nervous system structures and components of the sensory system specifically are notable limitations of the current study, as is the absence of analysis of NMJ in limb muscles. Evaluation of these structures in conjunction with behavioral tongue function analyses, as well as determining whether tongue exercise paradigms exert appreciable effects on structures of etiological interest to dysfunction in PD, are of interest for future research. The current behavioral training study design, in which rats underwent

eight weeks of exposure to testing conditions prior to final testing, constitutes an experimental control condition applicable to an expanded study design in which groups receiving progressive tongue press exercise are compared to outcomes of groups experiencing 'non-exercise' conditions in parallel. In this way, the muscle biology outcomes reported in this study are highly representative of control conditions for future studies specifically designed to evaluate the impact of progressive tongue exercise on outcomes of the *Pink1 -/-* model of PD.

## Supporting information

**S1 Raw images.**
(PDF)

## Acknowledgments

We gratefully acknowledge Dr. Masatoshi Suzuki's provision of muscle tissue from a rat model of amyotrophic lateral sclerosis, which was used in this study as a biological staining control for NCAM expression. We appreciate the provision of surgically denervated muscle samples and expertise in muscle isolation provided by Dr. John Russell and Jared Cullen. We are grateful for the assistance of Lily N. Stalter of the UW Surgery Statistical and Analysis and Research Programming (STARP) Core for assistance and advice regarding statistical analysis of temporal behavioral tongue press data. The D18 antibody developed by J.R. Sanes was obtained from the Developmental Studies Hybridoma Bank, created by the NICHD of the NIH and maintained at The University of Iowa, Department of Biology, Iowa City, IA 52242.

## Author Contributions

**Conceptualization:** Nadine P. Connor, Michelle R. Ciucci.

**Data curation:** Tiffany J. Glass, John C. Szot.

**Formal analysis:** Tiffany J. Glass, Cynthia A. Kelm-Nelson, John C. Szot.

**Funding acquisition:** Cynthia A. Kelm-Nelson, Nadine P. Connor, Michelle R. Ciucci.

**Investigation:** Tiffany J. Glass, Cynthia A. Kelm-Nelson, John C. Szot, Jacob M. Lake.

**Methodology:** Tiffany J. Glass, Cynthia A. Kelm-Nelson, John C. Szot, Jacob M. Lake.

**Project administration:** Nadine P. Connor, Michelle R. Ciucci.

**Resources:** Nadine P. Connor, Michelle R. Ciucci.

**Supervision:** Nadine P. Connor, Michelle R. Ciucci.

**Validation:** Tiffany J. Glass.

**Writing – original draft:** Tiffany J. Glass, Cynthia A. Kelm-Nelson.

**Writing – review & editing:** Tiffany J. Glass, Cynthia A. Kelm-Nelson, Nadine P. Connor, Michelle R. Ciucci.

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
