## [Decision Letter · Decision Letter 0]

17 Jun 2020

PONE-D-20-06880

Functional characterization of extrinsic tongue muscles in the Pink1-/- rat model of Parkinson disease

PLOS ONE

Dear Dr. Glass,

Thank you for submitting your manuscript to PLOS ONE. After careful consideration, we feel that it has merit but does not fully meet PLOS ONE’s publication criteria as it currently stands. Therefore, we invite you to submit a revised version of the manuscript that addresses the points raised during the review process.

We look forward to receiving your revised manuscript.

Kind regards,

Sheila M Fleming, Ph.D

Academic Editor

PLOS ONE

Journal Requirements:

2. As part of your revision, please complete and submit a copy of the ARRIVE Guidelines checklist, a document that aims to improve experimental reporting and reproducibility of animal studies for purposes of post-publication data analysis and reproducibility: https://www.nc3rs.org.uk/arrive-guidelines. Please include your completed checklist as a Supporting Information file. Note that if your paper is accepted for publication, this checklist will be published as part of your article."

3. Thank you for inlcuding your ethics statement; "All activities involving rats were approved by the University of Wisconsin-Madison School of Medicine and Public Health Institutional Animal Care and Use Committee (IACUC) (Protocol numbers: M005486 and M005177). "

Please amend your current ethics statement to confirm that your named ethics committee specifically approved this study.

For additional information about PLOS ONE submissions requirements for ethics oversight of animal work, please refer to http://journals.plos.org/plosone/s/submission-guidelines#loc-animal-research  

Reviewers' comments:

Reviewer's Responses to Questions

**Comments to the Author**

1. Is the manuscript technically sound, and do the data support the conclusions?

Reviewer #1: Yes

Reviewer #2: Yes

2. Has the statistical analysis been performed appropriately and rigorously? 

Reviewer #1: Yes

Reviewer #2: Yes

3. Have the authors made all data underlying the findings in their manuscript fully available?

Reviewer #1: No

Reviewer #2: Yes

4. Is the manuscript presented in an intelligible fashion and written in standard English?

Reviewer #1: Yes

Reviewer #2: Yes

5. Review Comments to the Author

Reviewer #1: Thank you for the opportunity to review this manuscript investigating tongue force in relation to neuromuscular changes in the genioglossus and styloglossus within a Pink1-/- rat model of early stage Parkinson's disease. This study is of clinical importance, as a clear understanding of early pathological changes in oromotor behaviors secondary to PD may help determine the optimal stage to initiate medical intervention. This manuscript is well-written with considerable rigour.

Ethics statement, including IACUC institution and protocol numbers, is provided. The article does not fully explain the extent to which it will adhere to standards for data availability as authors note "some restrictions will apply".

Title and Abstract: Clear and concise summary. I suggest more strongly emphasizing that the conclusions in the abstract (i.e., "Findings do not indicate overt neuromuscular pathology...") are within the context of early disease stage and wide variability.

Introduction (Background and Objectives): Authors provide sufficient scienific background and relevant references, including the appropriateness of the Pink1-/- model in relation to human disease particulary for early onset PD and oromotor dysfunction. The expectation of an overshoot in force generation as a sign of pathology was not well described especially considering the cited literature of human and rat models have shown neuromuscular changes occuring with a reduction in function (i.e., reduced mastication rates and vocalization changes) as well as in post-mortem/late stage PD as opposed to early stages. Please clarify that increased tongue force was expected and the rationale for such an overshoot to be indicative of pathological change and/or compensation for other sensorimotor disruptions.

Experimental Design and Statistical Analyses: The description of study procedures was thorough, the majority of which are established within the literature.

I would suggest providing a figure/flowchart illustrating the timeline of events to clarify the duration of training periods, timing of surgical denervation as well as intervals of testing in relation to rat age. It seems that all rats completed tonge press measurements at 4 months; however, please clarify (page 6, line 129) as it is unclear how many completed tongue press measurements at each endpoint (i.e., the "the 12th and final week" wording sounds like the same timepoint).

Please explain how the number of rats was arrived at, any sample size calculations used, and the reason for unbalanced numbers (i.e., 7-8) between genotypes. It is unclear how "tongue press timing" was included as a variable and what role it served in relation to the research aims (page 6 line 137).

Please state if the data met assumptions for the ANOVA models and/or how assumptions were addressed.

It would be interesting to explore correlations among myofiber transition (SG) and total a-synuclein (GG) with the absolute tongue force and variability within the PINK1-/- rat model, which may contribute to the discussion. The authors may also considered developing and analyzing a specific metric of "fatigue/endurance" from the present data such as that seen in human PD subjects (Solomon et al., 2000).

Discussion:

Consider revising the first paragraph to be more concise, less description of prior work regarding late disease effects, and focus on the novel findings of the present study. Please add hypotheses for finding differential effects (myofiber transition vs. total a-synuclein) between the two specific muscles, GG and SG.

Since prior research reported a reduction in mastication rates and phonation behaviors were associated with neuromuscular change, could the overshoot in force be a demonstration of compensation for underlying incoordination/impaired sensorimotor integration?

Please discuss if the 12 (+/- 8) weeks of tongue press training and use of a low level of resistance (.2 g) potentially served as exercise (i.e., with more presses (or higher rates) at higher initial forces generated by the PINK genotype compared to WT--not only during training, but also potentially generalized to nutritive presses). Could an increase in force and licking rates serve as exercises for PINK rats that could have protective qualities against neuromuscular pathology and result in greater similiarity in muscle biology between genotypes?

Discussion lacks any mention of limitations. It largely neglects describing the clinical implications and specifically delineating what is left to be explored and next study designs.

Reviewer #2: This is solid work characterizing early neuromuscular correlates of oromotor alterations in the Pink1 rat model of PD. It follows from the group's previous work demonstrating oromotor alterations in the model in older rats. The authors' conclusion that sensorimotor alterations likely account for the greater tongue forces and force variability is likely true, given the lack of neuromuscular phenotype at the age of these rats.

The authors speculate that fatigue accounts for the within-session decreases in tongue speed and tongue force. Despite the significant interaction between genotype and timepoint for the speed measure, an alternative explanation could be that the Pink1 rats became satiated due to greater consumption during the early minutes of the task. The authors do not show parallel maximum force data as a function of time within-session, nor do they statistically analyze this measure. These data should be included and could shed light on the fatigue hypothesis, especially given the lack of supporting NMJ data.

A less likely hypothesis for the greater within-session slowing of tongue movements involves a dopamine mechanism. Fowler & Das reported within-session slowing of licking when rats were administered haloperidol. It is unclear whether the Pink1 rats in this study exhibit loss of dopamine at this age, but there could be alterations in dopamine release dynamics.

A NMJ denervation phenotype has been reported for Pink1 knockout animals. Assuming the authors did not quantify NMJs in spinal muscles, is the time course for bulbar vs spinal denervation known in the model?

6. PLOS authors have the option to publish the peer review history of their article (what does this mean?). If published, this will include your full peer review and any attached files.

Reviewer #1: No

Reviewer #2: No

---

## [Author Response · Author response to Decision Letter 0]

5 Sep 2020

Reviewers' comments:

Reviewer's Responses to Questions

Comments to the Author

1. Is the manuscript technically sound, and do the data support the conclusions?

Reviewer #1: Yes

Reviewer #2: Yes

2. Has the statistical analysis been performed appropriately and rigorously? 

Reviewer #1: Yes

Reviewer #2: Yes

3. Have the authors made all data underlying the findings in their manuscript fully available?

Reviewer #1: No

Reviewer #2: Yes

4. Is the manuscript presented in an intelligible fashion and written in standard English?

Reviewer #1: Yes

Reviewer #2: Yes

5. Review Comments to the Author

Reviewer #1: Thank you for the opportunity to review this manuscript investigating tongue force in relation to neuromuscular changes in the genioglossus and styloglossus within a Pink1-/- rat model of early stage Parkinson's disease. This study is of clinical importance, as a clear understanding of early pathological changes in oromotor behaviors secondary to PD may help determine the optimal stage to initiate medical intervention. This manuscript is well-written with considerable rigour.

Ethics statement, including IACUC institution and protocol numbers, is provided. The article does not fully explain the extent to which it will adhere to standards for data availability as authors note "some restrictions will apply".

Response: This has been edited to “No restrictions will apply”; a new S1 figure has been added which contains raw gels and blots, and all data is available upon request. 

Title and Abstract: Clear and concise summary. I suggest more strongly emphasizing that the conclusions in the abstract (i.e., "Findings do not indicate overt neuromuscular pathology...") are within the context of early disease stage and wide variability.

Response: The Abstract has been edited slightly to emphasize the early disease timepoints of the study.

Introduction (Background and Objectives): Authors provide sufficient scienific background and relevant references, including the appropriateness of the Pink1-/- model in relation to human disease particulary for early onset PD and oromotor dysfunction. The expectation of an overshoot in force generation as a sign of pathology was not well described especially considering the cited literature of human and rat models have shown neuromuscular changes occuring with a reduction in function (i.e., reduced mastication rates and vocalization changes) as well as in post-mortem/late stage PD as opposed to early stages. Please clarify that increased tongue force was expected and the rationale for such an overshoot to be indicative of pathological change and/or compensation for other sensorimotor disruptions.

Response: These are good points. The anticipated overshoot in force has been more clearly described in the Introduction.

Experimental Design and Statistical Analyses: The description of study procedures was thorough, the majority of which are established within the literature.

I would suggest providing a figure/flowchart illustrating the timeline of events to clarify the duration of training periods, timing of surgical denervation as well as intervals of testing in relation to rat age. 

It seems that all rats completed tonge press measurements at 4 months; however, please clarify (page 6, line 129) as it is unclear how many completed tongue press measurements at each endpoint (i.e., the "the 12th and final week" wording sounds like the same timepoint).

Response: A new flowchart (new Figure 1) has been introduced to describe the timeline of tongue training periods, and the accompanying Methods text has been updated to explain that the conclusion of the behavioral testing and euthanasia coincided with 4 months of age for one age cohort, and 6 months of age for another age cohort. The Methods has been edited to more clearly explain that surgically denervated biological control muscles for NMJ staining were obtained from WT rats that were not involved in any of the other experiments that incorporated Pink1 -/- rats.

Please explain how the number of rats was arrived at, any sample size calculations used, and the reason for unbalanced numbers (i.e., 7-8) between genotypes. 

Response: The Methods section has been edited to include this information. A typo in the original group size statement has been corrected; both age groups comprised of 7 to 8 rats per group.

It is unclear how "tongue press timing" was included as a variable and what role it served in relation to the research aims (page 6 line 137).

Response: Additional information has been added to the Methods section to describe data collection and analysis for tongue press timing. Additional information has been added to the Introduction to reflect the relevance of timing to disease manifestation in PD.

Please state if the data met assumptions for the ANOVA models and/or how assumptions were addressed.

Response: New information has been added to the ‘Methods: Statistics” section stating that the Shapiro-Wilk test was used to test the assumption of normality, and the Levene’s test for equality of variances. 

It would be interesting to explore correlations among myofiber transition (SG) and total a-synuclein (GG) with the absolute tongue force and variability within the PINK1-/- rat model, which may contribute to the discussion. 

Response: We agree that this would be an interesting correlation. We have correlated behavioral findings with neuropathology in other work, but this has also been criticism of our work because the correlation does not substantiate a mechanism.

The authors may also considered developing and analyzing a specific metric of "fatigue/endurance" from the present data such as that seen in human PD subjects (Solomon et al., 2000).

Response: We are grateful for the suggestion to consider Solomon et al., 2000, and have updated the Discussion section to acknowledge the intellectual contributions of Solomon to this area of research. In “Strength, Endurance, and Stability in Parkinson disease”, Solomon et al used IOPI to assess tongue strength by asking participants to “squeeze as hard as you can”, and tongue endurance was assessed by having the participant maintain 50% maximal pressure for as long as possible. It is possible an analogous metric could be developed in the rat tongue press paradigm which analyzes the duration of the test session though which 50% maximal tongue pressure is exerted, and this may be of interest for future work. However, the rat tongue press paradigm in this current manuscript has a concurrent timing component of the task, in the form of dynamic lick rates, which complicates the comparison. 

Discussion:

Consider revising the first paragraph to be more concise, less description of prior work regarding late disease effects, and focus on the novel findings of the present study. Please add hypotheses for finding differential effects (myofiber transition vs. total a-synuclein) between the two specific muscles, GG and SG.

Response: The first paragraph of the discussion has been edited to omit some description of prior work. This paragraph has also been adjusted to mention that differences in the degree to which genotype impacted GG and SG were not originally expected. Considerations for interpreting levels of a-synuclein in these two muscles are discussed in the second paragraph of the discussion.

Since prior research reported a reduction in mastication rates and phonation behaviors were associated with neuromuscular change, could the overshoot in force be a demonstration of compensation for underlying incoordination/impaired sensorimotor integration?

Response: This is an interesting question. While direct measures of incoordination or impaired sensorimotor integration are beyond the scope of this study, the second-to-last paragraph of the Discussion raises these possibilities.

Please discuss if the 12 (+/- 8) weeks of tongue press training and use of a low level of resistance (.2 g) potentially served as exercise (i.e., with more presses (or higher rates) at higher initial forces generated by the PINK genotype compared to WT--not only during training, but also potentially generalized to nutritive presses). 

Response: The language in the Methods section has been adjusted to clarify that .2g of force is a trivial amount of force that is applied as a consequence of merely licking water which has been dispensed onto the disk. Since rats under typical, experimentally naive conditions also obtain all of their water intake through licking, .2g in this experimental context is regarded to be a ‘control’ or ‘non-intervention’ licking condition, and not a true ‘press’ condition. We have also further clarified that rats during the 8-week session were removed from the testing apparatus after initial demonstration of compliance with the lick procedure, to further eliminate possibilities of unintended training effects on the tongue muscles. 

Could an increase in force and licking rates serve as exercises for PINK rats that could have protective qualities against neuromuscular pathology and result in greater similiarity in muscle biology between genotypes?

Response: To address this question the last paragraph of the Discussion section has been updated to explain that the tongue press paradigms employed here were construed as behavioral tests of function within the broader context of ‘control’ / ‘non-exercise’ conditions. The Methods text and new accompanying figure have been clarified to explain that the eight-week interval of licking did not demand appreciable or increasing forces. The 3-day test sessions which elicit increased tongue press forces are not believed to have entailed sufficient duration of exercise for appreciable impacts on the muscle biology measures studied. Experimental paradigms designed to elicit exercise-induced changes in tongue muscle biology are commonly eight weeks of progressive demands for increased tongue forces.

Discussion lacks any mention of limitations. It largely neglects describing the clinical implications and specifically delineating what is left to be explored and next study designs.

Response: Although the discussion does not have a ‘limitations section’, it has now been expanded to incorporate the limitations throughout the discussion. The discussion has been edited to reflect that the absence of concurrent evaluation of central nervous system structures and components of the sensory system specifically are notable limitations. Finally, the last paragraph of the discussion describes future study designs.

Reviewer #2: This is solid work characterizing early neuromuscular correlates of oromotor alterations in the Pink1 rat model of PD. It follows from the group's previous work demonstrating oromotor alterations in the model in older rats. The authors' conclusion that sensorimotor alterations likely account for the greater tongue forces and force variability is likely true, given the lack of neuromuscular phenotype at the age of these rats.

The authors speculate that fatigue accounts for the within-session decreases in tongue speed and tongue force. Despite the significant interaction between genotype and timepoint for the speed measure, an alternative explanation could be that the Pink1 rats became satiated due to greater consumption during the early minutes of the task. 

Response: The possibility of satiation is believed to be unlikely due to our observations that rats consumed ad libitum water provided at the conclusion of testing sessions. Unfortunately, these volumes of water consumed were not analyzed as part of the present study. 

The authors do not show parallel maximum force data as a function of time within-session, nor do they statistically analyze this measure. These data should be included and could shed light on the fatigue hypothesis, especially given the lack of supporting NMJ data.

Response: These new data and analyses have been included in Figure 2. The Methods: Statistics section has been updated with new analyses of tongue press data as a function of time within-session.

A less likely hypothesis for the greater within-session slowing of tongue movements involves a dopamine mechanism. Fowler & Das reported within-session slowing of licking when rats were administered haloperidol. It is unclear whether the Pink1 rats in this study exhibit loss of dopamine at this age, but there could be alterations in dopamine release dynamics.

Response: Thank you for that comment. The discussion section has been updated to note that our previous work with haloperidol (Ciucci & Connor, 2009) and 6-OHDA (Ciucci et al, 2011) shows that blocking D2 dopamine receptors and lesioning the medial forebrain bundle unilaterally both lead to decreased tongue forces and lick rates. However, in this model, at the specified ages (4, 6 months) our results agree with Dave, et al., at 8 months (Grant et al., 2015) that Pink1-/- rats do not show significant dopamine depletion. Alternatively, our studies within this model show: reduced TH staining in the locus coeruleus, as well as increased alpha synuclein in areas that modulate dopaminergic and noradrenergic-mediated brainstem structures. 

A NMJ denervation phenotype has been reported for Pink1 knockout animals. Assuming the authors did not quantify NMJs in spinal muscles, is the time course for bulbar vs spinal denervation known in the model?

 Response: The Discussion has been updated to mention denervation phenotypes in Pink1 knockout mice. To our knowledge, the time course of bulbar vs spinal denervation has not been studied in Pink1-/- rats.

6. PLOS authors have the option to publish the peer review history of their article (what does this mean?). If published, this will include your full peer review and any attached files.

Do you want your identity to be public for this peer review? For information about this choice, including consent withdrawal, please see our Privacy Policy.

Reviewer #1: No

Reviewer #2: No

---

## [Decision Letter · Decision Letter 1]

25 Sep 2020

Functional characterization of extrinsic tongue muscles in the Pink1-/- rat model of Parkinson disease

PONE-D-20-06880R1

Dear Dr.Glass

We’re pleased to inform you that your manuscript has been judged scientifically suitable for publication and will be formally accepted for publication once it meets all outstanding technical requirements.

Kind regards,

Sheila M Fleming, Ph.D

Academic Editor

PLOS ONE

---

## [Editor Report · Acceptance letter]

1 Oct 2020

PONE-D-20-06880R1 

Functional characterization of extrinsic tongue muscles in the *Pink1-/-* rat model of Parkinson disease 

Dear Dr. Glass:

I'm pleased to inform you that your manuscript has been deemed suitable for publication in PLOS ONE. Congratulations! Your manuscript is now with our production department. 

Kind regards, 

on behalf of

Dr. Sheila M Fleming 

Academic Editor

PLOS ONE